# Characterization of Upper Extremity Kinematics Using Virtual Reality Movement Tasks and Wearable IMU Technology

**DOI:** 10.3390/s24010233

**Published:** 2023-12-30

**Authors:** Skyler A. Barclay, Lanna N. Klausing, Tessa M. Hill, Allison L. Kinney, Timothy Reissman, Megan E. Reissman

**Affiliations:** EMPOWER Laboratory, University of Dayton, Dayton, OH 45469, USAakinney2@udayton.edu (A.L.K.); treissman1@udayton.edu (T.R.); mreissman1@udayton.edu (M.E.R.)

**Keywords:** biomechanics, physical therapy, task-specific training

## Abstract

Task-specific training has been shown to be an effective neuromotor rehabilitation intervention, however, this repetitive approach is not always very engaging. Virtual reality (VR) systems are becoming increasingly popular in therapy due to their ability to encourage movement through customizable and immersive environments. Additionally, VR can allow for a standardization of tasks that is often lacking in upper extremity research. Here, 16 healthy participants performed upper extremity movement tasks synced to music, using a commercially available VR game known as Beat Saber. VR tasks were customized to characterize participants’ joint angles with respect to each task’s specified cardinal direction (inward, outward, upward, or downward) and relative task location (medial, lateral, high, and/or low). Movement levels were designed using three common therapeutic approaches: (1) one arm moving only (unilateral), (2) two arms moving in mirrored directions about the participant’s midline (mirrored), or (3) two arms moving in opposing directions about the participant’s midline (opposing). Movement was quantified using an XSens System, a wearable inertial measurement unit (IMU) technology. Results reveal a highly engaging and effective approach to quantifying movement strategies. Inward and outward (horizontal) tasks resulted in decreased wrist extension. Upward and downward (vertical) tasks resulted in increased shoulder flexion, wrist radial deviation, wrist ulnar deviation, and elbow flexion. Lastly, compared to opposing, mirrored, and unilateral movement levels often exaggerated joint angles. Virtual reality games, like Beat Saber, offer a repeatable and customizable upper extremity intervention that has the potential to increase motivation in therapeutic applications.

## 1. Introduction

Task-specific training has been shown to be an effective neuromotor rehabilitation intervention [1,2,3,4,5,6,7]. However, these tasks are often repetitive and encompass a combination of activities of daily living (ADL) that are difficult to quantify. Virtual reality (VR) has become increasingly popular in research due to its ability to present clearly defined movement tasks in realistic and engaging environments, while also recording quantitative information about the participants’ movements [8,9,10,11,12,13,14,15]. Utilizing customization capabilities within VR, it has been shown possible to provide highly engaging task-specific experiences that define and guide the movement task [16].

As advances in VR allow for the presentation of well-defined movement tasks, a variety of biosensors facilitate the collection of associated quantitative data about how the tasks were performed. Particularly for upper extremity (UE) movement, more quantitative kinematic assessments could be especially useful, as they have been less studied than qualitative, visual, and spatiotemporal metrics [17]. Advances in inertial measurement unit (IMU) sensors and their associated systems have been shown to provide high-quality quantitative data at a lower cost than other conventional methods, including camera-based motion capture [17,18,19,20,21,22]. Prior works have demonstrated the feasibility of using IMU systems to assess the clinic or home activity of the UE. For example, a small chronic stroke cohort demonstrated the feasibility of using the Xsens IMU system to quantify UE workspace, number of reaches, and joint kinematics at time points over several months [23]. As biosensors improve in capability, ease of use, portability, and cost, they become increasingly feasible for adoption and integration beyond research labs to therapeutic and clinical use [19,20,21,22,23]. However, data from these IMU systems would be more comparable and useful if also collected during specific, known UE tasks in addition to general activities of daily living.

In general, UE metrics and movements are complex and lack standardization [17]. Comparatively, lower extremity metrics and gait behaviors have been published in multiple normative databases and are extremely standardized [24]. ADLs are commonly used in UE assessment with qualitative, visual, and spatiotemporal metrics being the typical outcomes [25,26,27]. For example, a VR-based therapy intervention in a chronic stroke cohort presented UE tasks of hammering, pouring, catching, and reaching [15]. Participants rated high enjoyment and authors noted that mirrored tasks were particularly useful for this population. Post-therapy clinical tests showed improvement versus pre, but unfortunately, kinematic behaviors during the tests were not recorded. Another VR-based intervention in a chronic stroke cohort did demonstrate a kinematic range of motion improvements for certain joints and directions, pre vs. post, but did not collect kinematics during the therapy sessions [14]. Qualitative metrics such as surveys, timings, and observer ratings are often considered less accurate, as they can be aggregate metrics or are subjective to the participant and/or researcher. These metrics are also typically less sensitive to the quality of the movements which could be clinically relevant [17].

Our work proposes that by combining VR and IMU technologies cyclic and repetitive task-specific training of UE movement could be standardized and evaluated, in a similar manner as lower extremity gait analysis. Using VR, custom environment tasks could be designed based on three common UE therapeutic approaches: (1) one arm moving only (unilateral), (2) two arms moving in mirrored directions about the participant’s midline (mirrored), or (3) two arms moving in opposing directions about the participant’s midline (opposing). Previous UE studies have identified small differences between the effect unilateral and bilateral movement therapy have and their respective benefits to the brain, but they also highlight that more comparative studies are required to better solidify our understanding of the role parallel tasks play [28,29,30,31,32]. Following stroke, bilateral motions, compared to unilateral motions, have shown an increase in activity of the ipsilesional primary motor and premotor cortices and may be more effective at improving strength, movement quality, and distal function [28,31,32]. Within bilateral rehabilitation applications, the motions can be mirrored about the participants’ midline (mirrored) or opposing about the participants’ midline (opposing). Many studies have identified benefits to mirror therapy and mirrored motions for people who have had a stroke or have been diagnosed with Cerebral Palsy [15,33,34,35,36]. With the addition of a physical mirror, there is a visual and muscle symmetry that is translated to the brain. This tricks the brain into believing the participant is moving symmetrically and produces therapeutic benefits beyond the training session [36]. Even without the presence of a physical mirror, task-specific therapy of mirrored bilateral motions has been found to increase the benefits of therapy based on the principle that the symmetric motions will activate similar neural networks in both hemispheres [15,28,34].

Previously discussed studies have used task-specific training, VR environments and games, and/or IMU technology for analyzing movement in both the upper and lower extremities. Here we aim to show through the combination of all of these ideas that UE training can be designed such that it can yield detailed quantitative analyses of repeated movement patterns while remaining engaging. Specifically, we aim to use IMU technology to characterize how peak UE joint angles are impacted by the direction and location of movement tasks posed in a popular and customizable VR game known as Beat Saber. Additionally, we aim to determine if UE joint angles for a specific task are significantly impacted by the given parallel task condition: unilateral, mirrored, or opposing movement tasks. Based on the general kinematics of UE movement, it is hypothesized that horizontal tasks (inward and outward motions) and lateral targets will increase shoulder abduction, wrist flexion, and wrist extension joint angles. It is also hypothesized that vertical tasks (upward and downward motions) and high targets will increase shoulder flexion, wrist radial deviation, and wrist ulnar deviation joint angles. Lastly, it is hypothesized that opposing tasks will create the least range of motion due to increased task complexity.

## 2. Materials and Methods

### 2.1. Equipment/Software

The Vive Pro VR system (HTC, Taichung City, Taiwan) was used to create and immerse the participants in the virtual environment. The headset uses an AMOLED display with a 90 Hz refresh rate, a 110-degree field of view, and synchronized audio. The commercially available game, Beat Saber (Beat Games, v1.12.1, Prague, Czech Republic) presents songs with target blocks in specific placements on a grid and tasks the participant to move in a slicing motion through the middle of the block in a certain direction with a lightsaber-based on the song’s beats (the motion of the lightsaber reflects a 1-m vector extending from the end of the handheld controller and matching the handheld controller’s orientation). In addition to pre-defined levels, the software has a graphic user interface that allows full customization of user-defined levels. This includes any choice of song, target speed, target block placement in the grid, and target direction in which target blocks must be sliced in half with the motion of the lightsaber.

UE movement data were recorded using the Xsens MTw Awinda System (MVN version 2020.0.0, Enschede, Netherlands). The system included 17 IMUs placed on the head, arms, and legs according to the Xsens MVN User Manual [37]. Body measurements were taken with a measuring tape and inserted into the software to create an appropriately sized motion capture model. A brief calibration was then completed to ensure the accuracy of the model. Each participant was asked to stand in a neutral position with both arms straight to the side and head facing forward. Participants then walked in a straight line, turning around when the software indicated. After the calibration was completed, participants were asked to touch their hands together and then to their chest. If the avatar’s hands were passing either each other or the chest, then the calibration was performed again until intersegment behavior matched real-world conditions.

### 2.2. Description of Movement Levels and Movement Tasks

The customized Beat Saber movement levels used in this study were designed based on three common UE therapeutic approaches: unilateral, mirrored, and opposing. The three customized levels therefore distinguish the arm movements involved, which are referred to as the movement levels. Unilateral movements consisted of movements using one arm at a time. Mirrored movements consisted of both arms moving in mirroring directions about the participant’s midline (e.g., both arms moving up or both arms moving away from the midline). Finally, opposing movements consisted of both arms moving in opposing directions about the participant’s midline (e.g., one arm moving up and the opposite arm moving down or both arms moving to the left). Each target block defined a UE movement task in one of the four cardinal directions (indicated by an arrow on the target block) and was labeled as inward, outward, upward, or downward. Where inward movement was toward the midline of the body and outward movement was away from the midline of the body regardless of the arm used. All selected songs had a steady rate within the range of 119 to 139 beats per minute resulting in 1.7 to 2.1 s between each task. Movement tasks aligned with the beat of the music and were spaced out evenly with 4 musical beats in between tasks.

Movement tasks were mapped on a 2 × 4 grid, shown in Figure 1. For consistency, the center of the top row was aligned with each participant’s shoulder joint center. The red target blocks are the participant’s left-arm targets and the blue target blocks are the participant’s right-arm targets. No movement tasks required an arm to cross the midline. The color of the actual target blocks in the game follows this same color coding, red lightsaber and target blocks for the participant’s left hand and blue lightsaber and target blocks for the participant’s right hand. The top row was defined as high and the bottom row as low. The inner two columns were defined as medial and the outer columns as lateral. These experimental factors result in a variety of possible task combinations (e.g., right hand, medial and high position, inward direction task).

### 2.3. Data Collection Procedure

This study recruited 16 healthy participants (7 females and 9 males, aged 21.69 ± 0.6 years, height of 1.76 ± 0.06 m, weight of 73.30 ± 10.22 kg) without any UE injuries. All protocols were approved by the University of Dayton’s Institutional Review Board with written informed consent from each participant. Inclusion criteria included ages 18 to 85 years old and having visual acuity of at least 20/30 as assessed by a Snellen vision test. Exclusion criteria included that participants could not have any current health conditions or injuries that would impact their ability to stand or balance or have a history of an upper limb injury, upper limb pain, or adverse responses to VR.

A brief UE range of motion test was performed at the beginning and repeated upon completion of the movement trials. The participant then played a tutorial Beat Saber level to become familiar with the motions and visual instructions. Participants completed the tasks while in a standing position and were instructed to place their arms at their sides between each task.

Upon completion of the practice level, participants were then given their choice of song to accompany the randomized movement level (mirrored, opposing, and unilateral) and pattern (pattern of task combination). All three movement levels were repeated twice, resulting in six movement trials with 72–96 tasks each depending on the level. Within a movement level, the task combinations were repeated three times sequentially.

### 2.4. Data Analysis

Visual 3D (C-motion Inc., v2022.08.2, Germantown, MD, USA) and MATLAB (The MathWorks^®^, R2022a, Natick, MA, USA) were used to process and analyze all data including segment motion and joint angles for the wrist, elbow, and shoulder. XSens MVN files auto-generated a 15-segment model in Visual 3D for each participant. Given known task spacing based on the song’s beats per minute, task events at the correct frequency were generated and shifted in time across the data. Tasks occurred every four musical beats and will be referred to as “beat events”. For each time shift the total arm energy (potential and kinetic energy) was calculated across all beat events. The time shift that produced the greatest net arm energy total across all tasks was used as the time shift that defined beat event placement for all further analysis. With the timing of the beat events established in each dataset, beat profiles were defined as the movements performed in the 0.8 s preceding and following the beat event. The 0.8 s period was chosen as it represents the time between the beat and offbeat during the fastest song. Thus, all profiles represent the same amount of time and no time warping of the data occurs. Movement task definitions had three possible patterns for each level (e.g., unilateral 1, mirrored 3) and the patterns tested were recorded for each participant. Based on the known movement task patterns given, each beat profile in a trial was labeled according to hand, vertical position (high or low), horizontal position (medial or lateral), and cardinal movement task direction (inward, outward, upward, or downward). Metrics calculated include peak shoulder, elbow, and wrist joint angles, endpoint (hand) velocity in the torso coordinate system, and hand velocities both on- and off-axis. The elbow angle was defined as the absolute angle between the upper and lower arm, whereas the shoulder and wrist angles were taken for each biomechanical axis. On- and off-axis velocities were defined as the hand velocities parallel and perpendicular to the specified movement task direction. The velocities were taken at the peak hand velocity magnitude around the beat.

All statistical processing was performed with NCSS (Number Cruncher Statistical Systems, v2022, LLC. Kaysville, UT, USA). A repeated measures ANOVA was performed followed by Tukey–Kramer pairwise comparisons on any metrics with primary significance. Significance was set at *p* < 0.05. However, significant factors with Power Values (P) lower than 0.8 are not reported. Within factors included vertical position, horizontal position, and movement level. Left and right-side movements were grouped together, and direction (vertical, horizontal) was separated into two separate analyses. All peak joint angle values are represented in box plots as absolute values, even if they are negative.

## 3. Results

Shoulder, elbow, and wrist angle metrics, as well as hand velocity metrics, will be presented in the same structure. The time series trajectory of the metrics data will be separated into each cardinal movement task direction, movement level, and task position. The box plots are separated first by cardinal movement task direction, then by vertical and horizontal positions, as well as movement level.

Each box plot was created with at least 720 movements performed by the 16 participants. The * symbol above a box plot indicates a significant difference between the corresponding task position within a movement level (e.g., between high unilateral and low unilateral) and brackets indicate a significant difference between the movement levels.

### 3.1. Hand Velocity

The hand velocity magnitude trajectory during each beat profile is shown in Figure 2. Inward compared to outward cardinal movement task directions (horizontal movement tasks) and upward compared to downward cardinal movement task directions (vertical movement tasks) are near reflections. Vertical movement tasks had two distinct peaks whereas the horizontal movement tasks had three distinct peaks. The vertical position did not show a clear effect on hand velocity magnitude.

For both horizontal and vertical movement task directions, high targets had significantly faster peak hand velocity magnitudes compared to low targets for all movement levels (all *p* < 0.001, *p* > 0.85) [Figure 3]. For horizontal movement task directions, unilateral had significantly faster peak hand velocity magnitudes compared to mirrored and opposing (all *p* < 0.01, *p* > 0.85) [Figure 3, lower]. While not shown in the box plot, vertical movement tasks had significantly faster peak hand velocity magnitudes compared to horizontal movement tasks (*p* < 0.0001, *p* > 0.99). Vertical movements were on average 0.6 m/second faster than horizontal movements.

The on- and off-axis velocity trajectories during each beat profile are shown in Figure 4. Right around the beat, the on-axis velocity peaks and the off-axis crosses zero. For upward tasks, the on-axis velocity has a positive peak and downward tasks have a negative peak. The same is true for inward and outward, where inward has a negative peak and outward has a positive peak. For absolute hand velocity, in Figure 5, factors of cardinal movement direction type and hand velocity axis were found significant (all *p* < 0.0001, *p* > 0.99).

### 3.2. Joint Angles

As kinematic behaviors for movement tasks in different cardinal directions were anticipated to be different, statistical comparisons between these are not shown explicitly. The resultant plots and statistical differences in this study therefore focus on the factors of task position and movement level, due to responses to these factors being anticipated to be more nuanced.

The frontal shoulder abduction/adduction trajectories during each beat profile are shown in Figure 6. Trajectories highlight that upward and outward tasks result in the kinematic peak shoulder abduction occurring between 50% and 75% of the beat profile. Conversely, downward and inward tasks result in kinematic trajectories that roughly mirror the opposite direction task and have peak angles between 25% and 50% of the beat profile. Compared to medial targets, lateral targets consistently resulted in more exaggerated frontal plane shoulder behaviors. Inward compared to outward cardinal movement task directions (horizontal movement tasks) as well as upward compared to downward cardinal movement task directions (vertical movement tasks) are near reflections with the exception of the unilateral movement level. All profiles are bell-shaped with the horizontal movement tasks reaching a higher peak than the vertical movement tasks. Unilateral movements tended to have larger shoulder abduction and less consistency between cardinal movement task directions around the beat. Lateral targets tended to have more shoulder abduction overall.

For both horizontal and vertical movement tasks directions, high targets had significantly larger peak shoulder abduction angles compared to low targets for all movement levels (all *p* < 0.0001, *p* > 0.99) [Figure 7]. Also, for both horizontal and vertical movement task directions, lateral targets had significantly larger peak shoulder abduction angles compared to medial targets for all movement levels (all *p* < 0.0001, *p* > 0.99) [Figure 7]. For vertical movement task directions, unilateral tasks had significantly higher shoulder abduction than both other movement levels when the task placements were high or lateral (all *p* < 0.001) [Figure 7, upper]. Additionally, opposing tasks had significantly higher shoulder abduction than mirrored for high and lateral task placements (all *p* < 0.05). For horizontal movement task directions, the same relationships were observed across all task placements. Unilateral tasks had significantly higher shoulder abduction than both other movement levels (all *p* < 0.0001) and mirrored tasks were significantly higher than opposing (all *p* < 0.01) [Figure 7, lower]. While not shown in the box plot, horizontal movement tasks had significantly larger peak shoulder abduction angles compared to vertical movement tasks (*p* < 0.0001, *p* > 0.99). Horizontal movement tasks had on average 5 degrees more shoulder abduction than vertical movements.

The sagittal shoulder flexion/extension trajectories during each beat profile are shown in Figure 8 left column. Trajectories highlight that shoulder flexion tends to peak before the beat for downward tasks, after the beat for upward tasks, and roughly at the beat for inward and outward tasks (the beat being at 50%). However, all cardinal movement task directions still resulted in kinematic trajectories that are generally bell-shaped. Compared to low targets, high targets consistently resulted in more exaggerated sagittal plane shoulder behaviors.

For both horizontal and vertical movement task directions, high targets had significantly larger peak shoulder flexion angles compared to low targets for all movement levels (all *p* < 0001, *p* > 0.95) [Figure 9]. For vertical movement task directions, opposing tasks had significantly fewer shoulder flexion angles compared to mirrored and unilateral tasks (all *p* < 0.0001, *p* > 0.95) [Figure 9, upper].

The elbow flexion/extension trajectories during each beat profile are shown in Figure 8 right column. The trajectories highlight that for horizontal movement tasks the inward and outward cardinal movement task directions result in kinematic elbow patterns that are near reflections. The same is shown for vertical movement tasks, with upward and downward movement directions showing the same but reflected pattern. The peaks for inward and upward tasks occur around the same time after the beat compared to outward and downward tasks which both occur before the beat. Overall, vertical movement tasks demonstrate more exaggerated elbow flexion trajectories than horizontal movement tasks. For both horizontal and vertical movement task directions, high and medial targets had significantly more peak elbow flexion angles compared to low and lateral targets, respectively (all *p* < 0.01, *p* > 0.90) [Figure 10]. For all horizontal movement tasks, as well as high and medial vertical movement tasks, unilateral had significantly more peak elbow flexion angles compared to mirrored and opposing (all *p* < 0.05, *p* > 85) [Figure 10]. Additionally, for low vertical movement tasks, unilateral had significantly more peak elbow angles, followed by mirror, compared to opposing tasks (all *p* < 0.01, *p* > 0.85) [Figure 10, upper]. Also, for lateral vertical movement tasks, opposing had significantly fewer peak elbow flexion angles compared to unilateral and mirrored (all *p* < 0.001, *p* > 85) [Figure 10, upper]. While not shown in the box plot, vertical movement tasks had significantly more peak elbow flexion angles compared to horizontal movement tasks (*p* < 0.0001, *p* > 0.99). Vertical movement tasks had on average 20 degrees more elbow flexion than horizontal movement tasks.

The kinematic wrist trajectories during each beat profile are shown in Figure 11, with sagittal angles on the left and frontal angles on the right. The neutral wrist angle (determined at 0% of the beat profile) was roughly 10 degrees extension in the sagittal plane and 5 degrees ulnar deviation in the frontal plane. Unlike other joint kinematic trajectories, the wrist typically demonstrated joint angle deviations in both directions away from the neutral position. For example, in the sagittal plane, a more extended and a more flexed joint angle were both expressed during the beat profile. However, due to the non-zero neutral position, the wrist only enters true flexion for inward cardinal movement task directions. Similarly, in the frontal plane, radial deviation in the wrist only occurs during upward and downward cardinal movement task directions. Compared to the previous joint trajectories, each direction has a more unique pattern. All other directions stay in extension the entire time. Outward and upward cardinal movement task directions hit the peak wrist extension right after the beat. Inward and downward cardinal movement task directions hit the peak before the beat with inward going into flexion after the beat. Upward compared to downward cardinal movement task directions, vertical movement tasks, are near reflections. Frontal plane range of motion was more exaggerated for vertical movement tasks, compared to horizontal movement tasks. While sagittal range of motion was more exaggerated for horizontal movement tasks compared to vertical movement tasks.

For unilateral vertical movement task directions, lateral targets had significantly more sagittal wrist range of motion compared to medial targets (*p* < 0.0001 *p* > 0.99) [Figure 12, upper]. For unilateral horizontal movement task directions, low targets had significantly more sagittal wrist range of motion compared to high targets (*p* < 0.005, *p* > 0.80) [Figure 12, lower]. While not shown in the box plots, horizontal movement tasks had significantly more sagittal wrist range of motion compared to vertical movement tasks (*p* < 0.05, *p* > 0.90). Horizontal movements had on average 9 degrees greater range of motion in the sagittal wrist angle.

For mirrored vertical movement task directions, lateral targets had significantly more sagittal wrist range of motion compared to medial targets (*p* < 0.0001 *p* > 0.90) [Figure 13, upper]. For horizontal movement task directions, mirrored had significantly more sagittal wrist range of motion compared to opposing and unilateral (*p* < 0.01, *p* > 0.99) [Figure 13, lower]. While not shown in the box plots, vertical movement tasks had significantly more sagittal wrist range of motion compared to horizontal movement tasks (*p* < 0.001, *p* > 0.95). Vertical movements had on average 13 degrees more range of motion in the frontal wrist angle.

## 4. Discussion

Recent developments in sensors over the last decade have allowed continual advances in the application of VR and IMU systems aimed at improving understanding of human movements and how movement tasks are performed. Particularly there has been greater adoption of headset-based VR systems [10,11,12], the use of multiple IMUs and sensor fusion to examine joint angle kinematics [11,12,13,15,21,22,23], and the use of VR to present specific movement tasks [11,12,13]. The current study aimed to build on prior work by examining all of those features listed in a single study. In addition, we examined questions related to unilateral, mirrored, and opposing upper extremity tasks which have noted utility for therapeutic applications [15,31,32]. In this study, a healthy control population’s UE joint kinematics were evaluated using an IMU motion capture system during an engaging and fully immersive VR game. The easily repeatable and cyclic movement tasks presented in the game allowed for unilateral and bilateral tasks to be examined and facilitated an analysis more comparable to gait, which has a much more standardized analysis method than most UE studies.

Velocity results from the XSens IMU system allowed for validation of the task performance, specifically that participants maximized their hand velocity in the target movement task direction at or near the task, indicated by the song beat, while simultaneously holding their off-axis hand velocity near zero. Participants’ upper extremity limb segment orientation data from the IMU system allowed for kinematic calculations to characterize the UE joint angle motions, or UE joint workspace, for each movement task factor. Movement tasks were presented as target blocks that the participant was instructed to make a slicing motion through in the indicated arrow direction on each target block. Movement tasks were paced at consistent intervals with the task timing to make the slicing motion through the target block with the beat of the music. Horizontal tasks were found to increase shoulder abduction and wrist flexion, as well as decrease wrist extension. Vertical tasks were found to increase shoulder flexion, wrist radial deviation, wrist ulnar deviation, and elbow flexion. The further the target was from the shoulder joint center, the more shoulder flexion and abduction occurred, for all cardinal movement task directions. Mirrored and unilateral movement levels often exaggerated joint angles, while opposing movement levels muted the metrics. These characterizations are important to physical therapists since often the goal of task-specific training is to increase the range of motion for specific joints and body limb segments [14,15,23]. A summary of the specifications that lead to the largest motion for each metric can be seen in Figure 14.

### 4.1. Task Validation

Body-worn IMU sensor motion capture systems, such as the XSens system used in this study, represent a comparatively low-cost method of obtaining quantitative kinematic data in clinical environments [20,21,22,23]. Additionally, VR represents a method for presenting well-defined movement tasks that can be consistently repeated or systematically modified to support therapeutic movement goals [10,11,12,14,15]. As in this study, a current limitation is that those two systems may be challenging to integrate either temporally or spatially. Thus, the motion capture data can only validate the general concept of the movement task and is unable to address more detailed aspects of accuracy. However, from the on- and off-axis velocity plot, in Figure 4, it can be validated that the participants are able to, on average, complete the correct task. Right around the beat, the on-axis velocity peaks and the off-axis crosses zero indicating that the participant was able to isolate their hand movement in the correct axis. For vertical tasks, the on-axis velocity has a positive peak in the upward direction and a negative peak in the downward direction. The same is true for inward and outward cardinal movement tasks, where inward has a negative peak and outward has a positive peak. This indicates that upward compared to downward cardinal movement tasks, as well as inward compared to outward tasks, were completed on the same axis but in the opposite direction.

### 4.2. Impact of Movement Levels

The study demonstrated that even for a healthy control population, significant differences in UE kinematics are discernable due to the movement level type, specifically if a parallel task is required and if the parallel task requires mirrored or opposing patterns between the two limbs. The hypothesis that the peak joint angles would be reduced for the opposing tasks was supported for most frontal shoulder angles, for sagittal shoulder angles in vertical movements, and for all elbow angles. Other comparisons resulted in no significant differences between opposing and other movement levels with the exception of frontal wrist angle range of motion in horizontal movements.

Most notably, unilateral movement levels resulted in significantly higher hand velocities in horizontal movements, higher frontal shoulder angles for most movements, and higher elbow angles for most movements. Thus, with regard to recommendations for targeting increased range of motion in clinical populations, unilateral movement tasks would be suggested. The exaggerated kinematic responses to the unilateral tasks may reflect the fact that the participant has no inter-arm collision or timing constraints to consider during the movement.

However, the kinematic differences observed in the mirrored and opposing movement levels likely reflect the fact that these tasks require the brain to consider additional constraints such as timing and inter-limb collision that are also critical UE capabilities. Prior work has shown that the brain processes mirrored, opposing, and unilateral motions differently and that unilateral and bilateral tasks may generate unique therapeutic benefits [28,31,32]. This is possibly because mirrored motions use symmetric muscle activation, making it the simplest motion for the brain to process. Whereas during opposing motions the brain generates a unique muscle activation pattern for each side of the body. Along with differences in UE limb muscle activation, opposing movements create a torque on the torso due to rotational accelerations of the arm masses that do not cancel out. This forces core muscles in the torso to activate and stabilize the body during opposing movement in the UE. Finally, parallel tasks in the horizontal direction typically required participants to slightly modify their behavior compared to the unilateral movement levels so that the hands would not collide during tasks. For all these reasons opposing movement tasks appear to result in a strategy that favors reduced joint angle excursion at the shoulder and elbow, sometimes accompanied by increased joint angle range of motion at the wrist.

### 4.3. Cardinal Movement Task Direction

Vertical motions compared to horizontal motions yielded statistically significant differences in the results. Horizontal motions are completed on the horizontal axis and the main joint motions associated with the horizontal axis are the frontal shoulder and sagittal wrist. Horizontal motion tasks have on average 5 degrees more shoulder abduction and 9 degrees more sagittal wrist range of motion compared to vertical motion tasks.

Vertical motions are completed on the vertical axis. The main joint motions associated with the vertical axis are the sagittal shoulder, frontal wrist, and elbow. Vertical motion tasks have on average 20 degrees more elbow flexion, 13 degrees more frontal wrist angle range of motion, and 0.6 m/second faster hand velocity magnitude than horizontal motion tasks. Vertical motions would be expected to have less wrist flexion than horizontal motions. However, this is not true. Wrist flexion still occurs during vertical motions due to the natural pronation/supination that occurs when the hands are moving in a lower position.

The faster hand velocity for vertical motion tasks is most likely due to the difference in movements required for vertical motions compared to horizontal motions. Vertical motions require two separate movements, up then down. Horizontal motions require three separate movements, up, over, then down. Due to the simplicity of the vertical motion, the participant is able to speed up to a higher velocity. The possibility of the hands hitting each other with horizontal motion also plays a role in slowing down the hands at the participant’s midline for the task.

This study demonstrated that even healthy controls were less able to isolate their hand velocity during horizontal tasks, showing increased off-axis velocity, compared to during vertical tasks (Figure 5). The three distinct velocity peaks seen in Figure 4 indicate directional changes. To achieve the horizontal movement task the brain must reduce vertical motion to zero at the correct time and spatial level (height). Then the brain must generate an UE muscle activation that resists gravity while generating appropriate medial or lateral velocities at the hand. It is worth noting that some on-axis velocity values were close to zero m/s. This is most likely due to the player incorrectly identifying the cardinal movement task direction. This vertical isolation is also present in Figure 4, the off-axis velocity stays tight around zero for upward and downward cardinal movement tasks indicating isolation around the on-axis that is not represented in the inward and outward cardinal movement tasks (horizontal direction).

### 4.4. Task Position

Vertical position compared to horizontal position yielded statistically different results. The main joint motions associated with a change in horizontal position are the frontal shoulder and sagittal wrist. Moving the target laterally by 0.5 m results in an average of 10 degrees more shoulder abduction, and 5 degrees less elbow flexion.

The main joint motions associated with a change in the vertical position are the sagittal shoulder, elbow, and frontal wrist. Moving the target up 0.5 m results in an average of 6 degrees more shoulder abduction, 15 degrees more shoulder flexion, and 0.3 m per second faster hand velocity.

The further lateral a target is placed the more shoulder abduction is required. The decrease in elbow flexion for lateral tasks indicates that the shoulder is compensating for the necessary motion to slice through the target blocks. The higher up a target is placed the larger shoulder flexion and elbow flexion required to reach. Shoulder abduction is larger for higher targets than lower targets and shoulder flexion is larger for more lateral targets than medial targets. This could possibly be due to the shoulder naturally abducting as it goes into flexion and vice versa.

### 4.5. Collision Constraint and Controller Hand Position

Inter-hand collisions would be a constraint in many of the parallel task movements tested regardless of the hands being empty or holding controllers. However, the fact of holding the VR controllers, which must be held in a certain position and which extend beyond the hands, adds an extra element to consider with regard to physical constraints on the movement. The physical constraints include a change in the baseline (or neutral) wrist angles, as well as possible collision of the controllers. Normal wrist notation indicates flexion-extension occurs on the vertical axis whereas radial and ulnar deviation occurs on the horizontal axis. The way that the controllers are held, shown in Figure 1, causes the notation to be flipped for this given study. The controllers also cause a constant state of flexion and ulnar deviation, creating a more flexed and ulnar-deviated neutral wrist position which can be seen in Figure 11. This indicates that the controllers have an influence on the baseline wrist joint metrics.

The collision constraint is present in peak hand velocity magnitude and sagittal wrist range of motion. Hand velocity is slower for horizontal motion tasks indicating that participants moved with caution so they would not crash the controllers together at the beginning or end of the motion. During horizontal motions, the sagittal wrist range of motion is also significantly smaller for unilateral motions. During mirrored and opposing motions, the controllers are at risk of colliding. Participants used more wrist motion in order to avoid any collision. Whereas with unilateral motions the participants can use the full range of their arms without any risk.

### 4.6. Limitations and Future Work

This study utilized body-worn IMU sensor motion capture technology (XSens). The use of IMU sensors to provide quantitative data in clinical and world environments is of great interest due to their low cost and small size [23]. Although more affordable and feasible for therapeutic translation, IMU sensors have limitations in how the data can be interpreted and their lower accuracy compared to camera-based IR motion capture systems [20,21,22]. Thus, we propose that future work would examine this VR-based protocol using other sensors including infrared (IR) motion capture and integrated VR (HTC Vive) tracker-based motion capture [38].

The current study was completed on young, healthy participants. This was necessary for characterizing how joint metrics change for the given factors. However, it does provide a baseline for comparison with movement-impaired populations. Multiple studies have found that similar methods of therapy could be useful in certain impaired populations such as stroke and cerebral palsy [7,9,12,23,31,32]. Therefore, future work aims to evaluate populations with movement impairments to characterize the peak biomechanical joint angle data and SPM differences with respect to those clinical populations.

As noted in the discussion, the IMU motion capture system was not synced directly to the VR environment. Thus, it was not possible to evaluate the exact timing or spatial aspects of the performance. In this study, our focus is limited to accessing the joint kinematic profiles expressed by healthy controls in response to movement targets presented in VR with known timing and task factors. The cyclic nature and known pacing of the tasks reduce the impact of this limitation and reflect a likely constraint of the co-use of VR and IMU systems in a clinical environment.

The benefit of the Beat Saber VR therapy is the gaming-like engagement coupled with the repetitive and cyclic motions that can be analyzed. The current statistical analysis is only on peak biomechanical joint angle data. However, in the future statistical parametric mapping (SPM) analyses will be used on the full-time series data [39]. SPM allows full movement profiles to be statistically compared for differences, rather than just peak values so that regions that differ can be identified. It should be noted that SPM is mainly used to evaluate gait and can be hard to use for UE motions [40,41]. UE motions do not typically have a cyclic motion with a clear beginning and end as we have in this study. Thus, additional analytical techniques with SPM like nonlinear registration can be used to adjust for any factor of early or late target movements [42].

Improvements in the sensors and systems used by VR trackers also suggest possible low-cost, quantitative motion capture methods. These would have the advantage of already being integrated with the VR systems and allow movement data to be synced to the VR environment. After validating VR tracker-based motion capture with IMU or IR-based motion capture systems, it is possible that only VR sensors will be necessary for analysis [21,38]. Thus, VR tracker-based motion capture could create a similar affordable and feasible option to what was valued in the IMU sensors.

## 5. Conclusions

As VR technology improves it presents a low-cost, engaging option for presenting different tasks and environments and studying human movements. There is a particular opportunity for standardizing and studying UE movements which are critical for activities of daily living but for which research is far less standardized versus lower extremity movements. Therapeutic applications of VR can benefit from popular yet customizable games like Beat Saber, which allow for fully engaging, motivating, and repeatable tasks for the UE. Wearable IMU technology, like XSens, additionally provides a low-cost solution to fully analyze such motions and produce quantitative kinematic data. In order to utilize such games for therapy and make any comparisons, a healthy cohort must be characterized first.

This study demonstrates that the combination of VR and IMU technology can be used to extensively quantify a cohort’s UE kinematics during the VR intervention. Beyond this, customized movement tasks can examine how factors of the movement task impact kinematic profiles and peak joint angles. From the current results, there is a potential for translational VR therapeutic approaches that could be recommended. For example, if a patient needs to work on their range of motion for shoulder abduction/adduction and wrist extension/flexion, then it can be suggested to provide horizontal motion tasks and lateral targets. Likewise, if a patient needs to work on their range of motion for shoulder extension/flexion, wrist radial/ulnar deviation, and elbow extension/flexion, then it can be suggested to provide vertical motion tasks and higher targets. In terms of translating VR into therapeutic approaches, this study suggests that unilateral motion tasks result in the highest range of joint motion, followed by mirrored motions. However, opposing motions appear to require more effort in coordinating the movement of both limbs and timing of the execution for the parallel tasks, as well as maintaining a stable torso. While this study characterizes normative behavior for this protocol, further research with clinical populations is required to understand how a population of interest may respond to the same factors and what movement tasks would be most useful for therapeutic purposes.

## Figures and Tables

**Figure 1 sensors-24-00233-f001:**
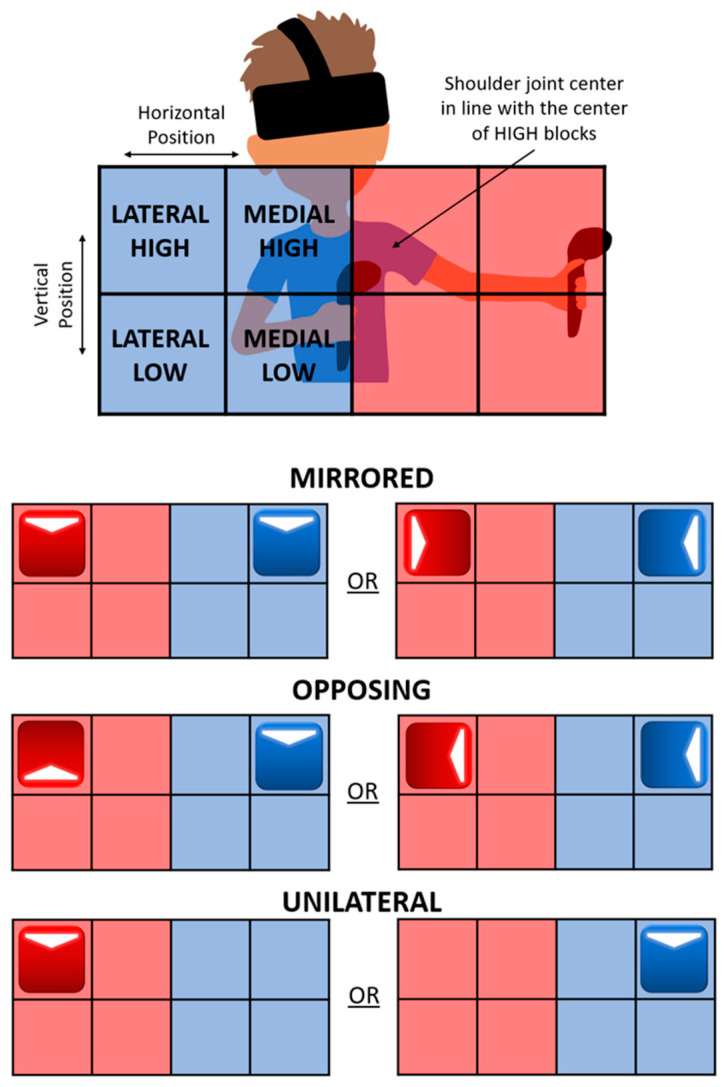
Approximate positioning and locations of block grid and movement level explanations. Arrows indicate task’s target position(s) and cardinal movement direction(s). Red indicates a left arm task and blue indicates a right arm task.

**Figure 2 sensors-24-00233-f002:**
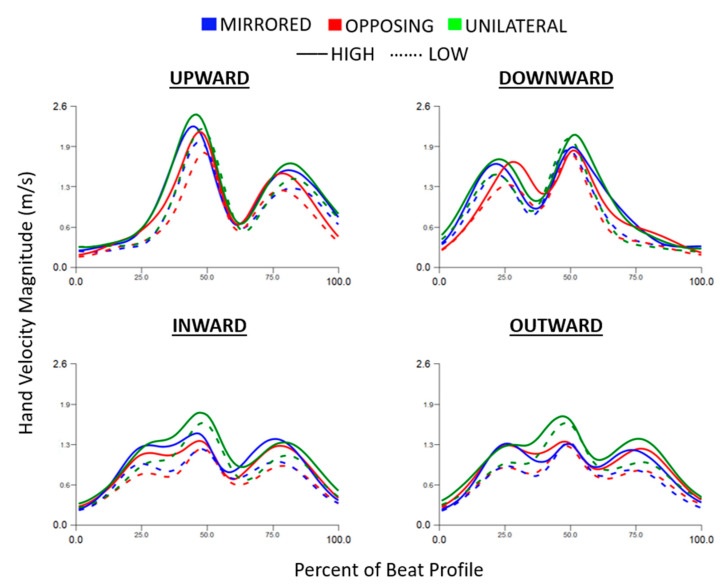
Hand velocity magnitude trajectory normalized to the beat profile and separated into each cardinal movement task direction. Movement levels and vertical task position are shown by line color and line style, respectively.

**Figure 3 sensors-24-00233-f003:**
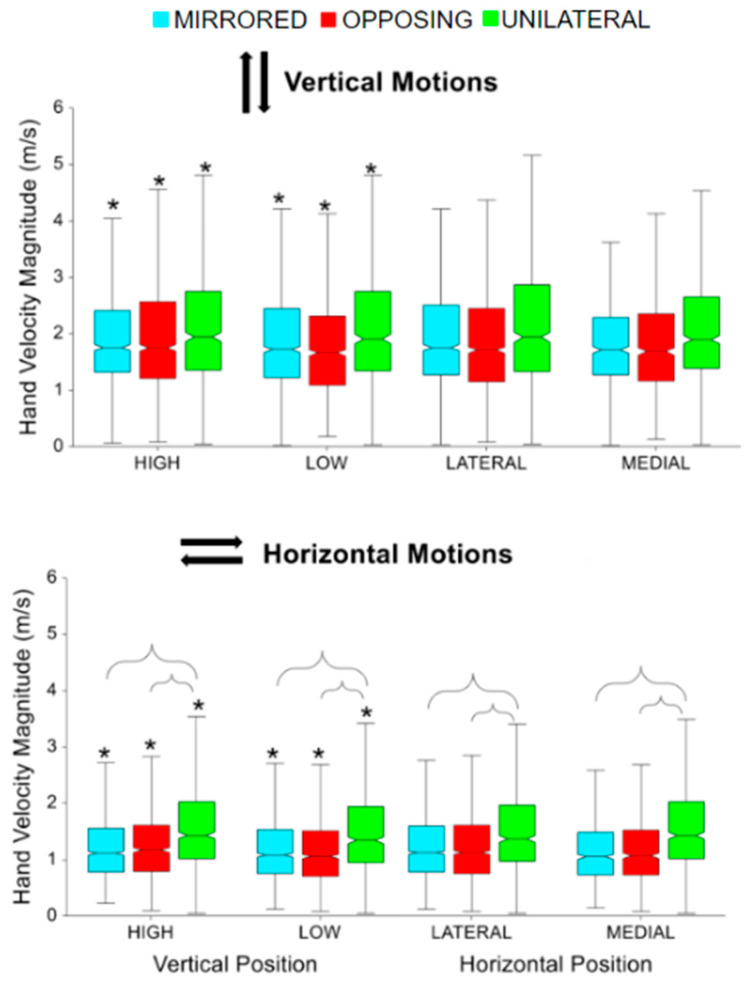
Peak hand velocity magnitude for vertical and horizontal movements comparing task position and movement level. The * symbol indicates a significant difference between metrics due to the vertical or horizontal within a movement level, e.g., between unilateral high and unilateral low. Brackets indicate a significant difference between movement levels, e.g., between unilateral high and mirrored high.

**Figure 4 sensors-24-00233-f004:**
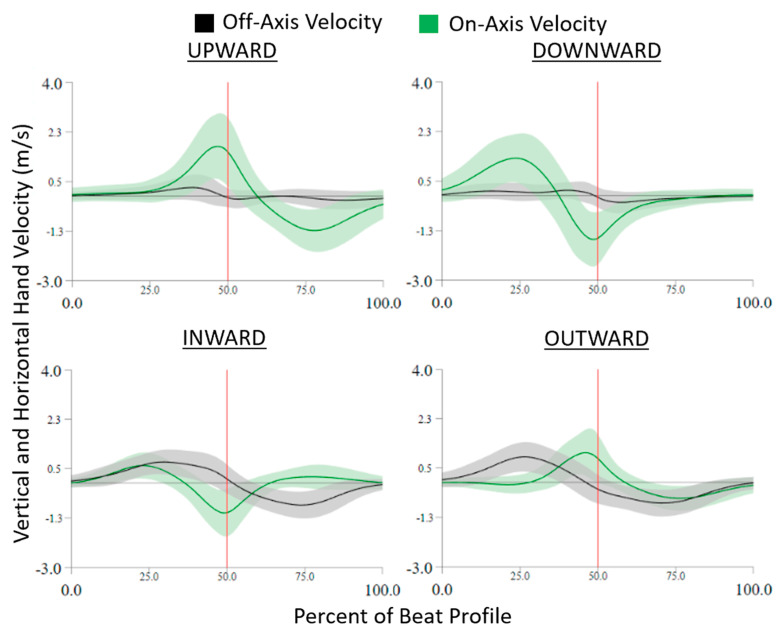
On− and off-axis velocity trajectories normalized to the beat profile and separated by cardinal movement task direction. On-axis velocity is represented by the green line, off-axis velocity is the black line, and the vertical red line indicates the beat location.

**Figure 5 sensors-24-00233-f005:**
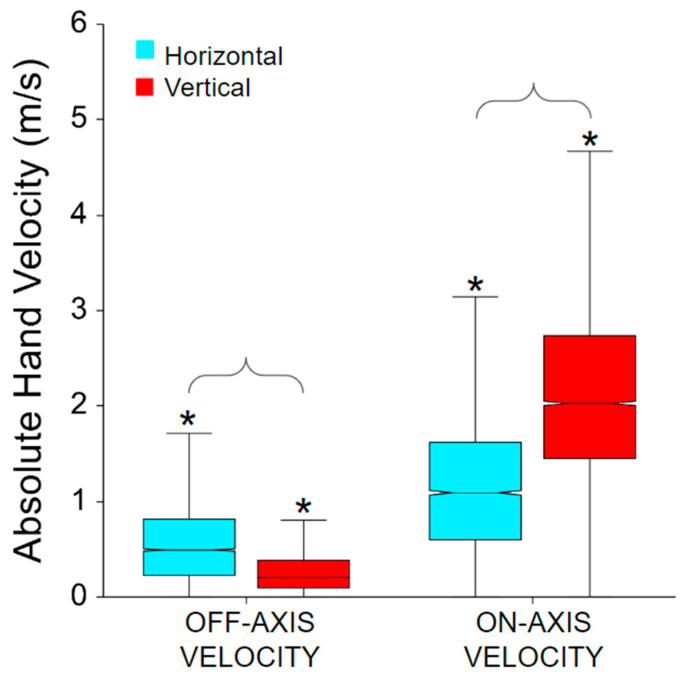
Absolute hand velocity for on- and off-axis as well as horizontal and vertical movement task direction. All interactions are significant. The * symbol indicates a significant difference between on- and off-axis velocities within the same movement direction, e.g., between horizontal off-axis and on-axis velocities. Brackets indicate a significant difference between movement directions, e.g., between off-axis horizontal and off-axis vertical velocities.

**Figure 6 sensors-24-00233-f006:**
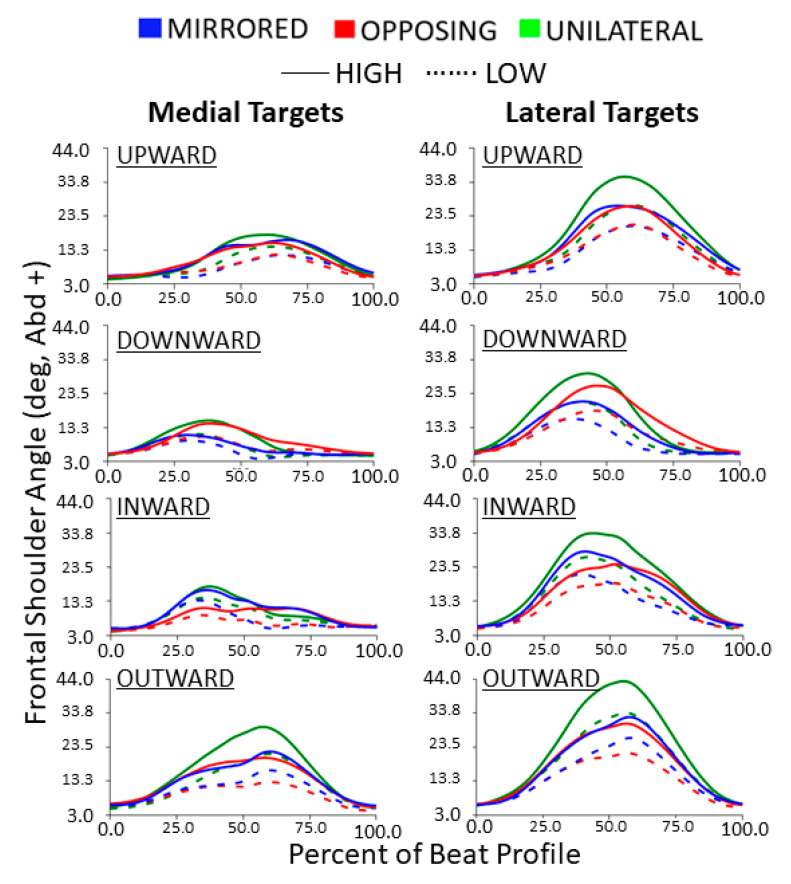
Frontal shoulder abduction/adduction joint angle trajectories normalized to the beat profile. Columns show medial and lateral targets. Rows show cardinal movement task direction. Movement levels and vertical task position are shown by line color and line style, respectively.

**Figure 7 sensors-24-00233-f007:**
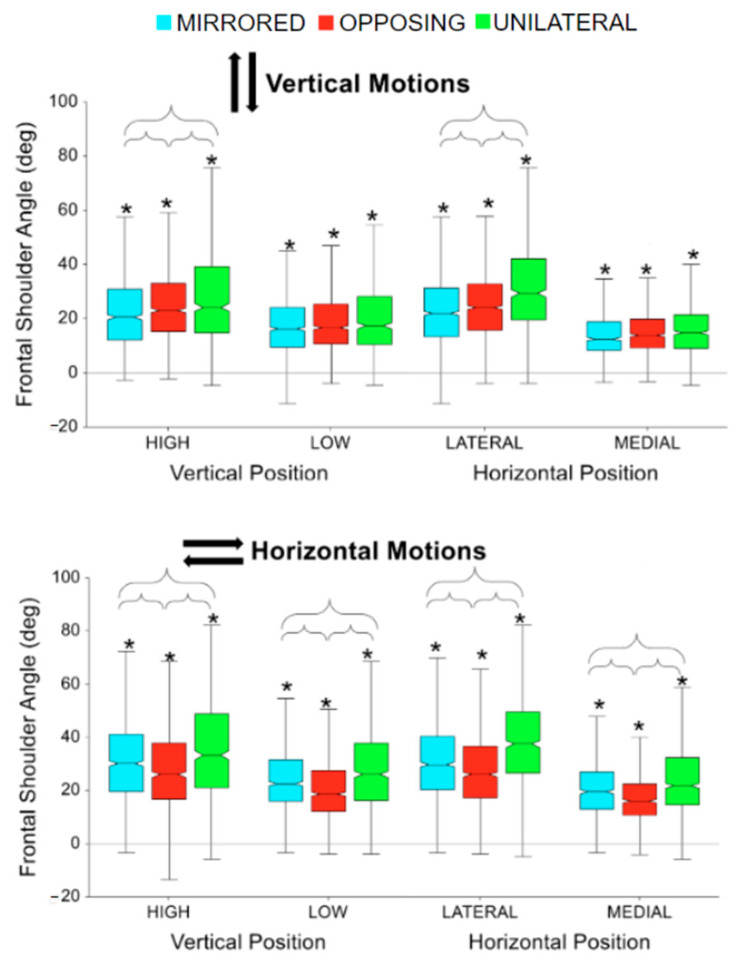
Peak shoulder abduction for vertical and horizontal movements comparing task position and movement level. The * symbol indicates a significant difference between metrics due to the vertical or horizontal position within a movement level, e.g., between unilateral high and unilateral low. Brackets indicate a significant difference between movement conditions, e.g., between unilateral high and mirrored high.

**Figure 8 sensors-24-00233-f008:**
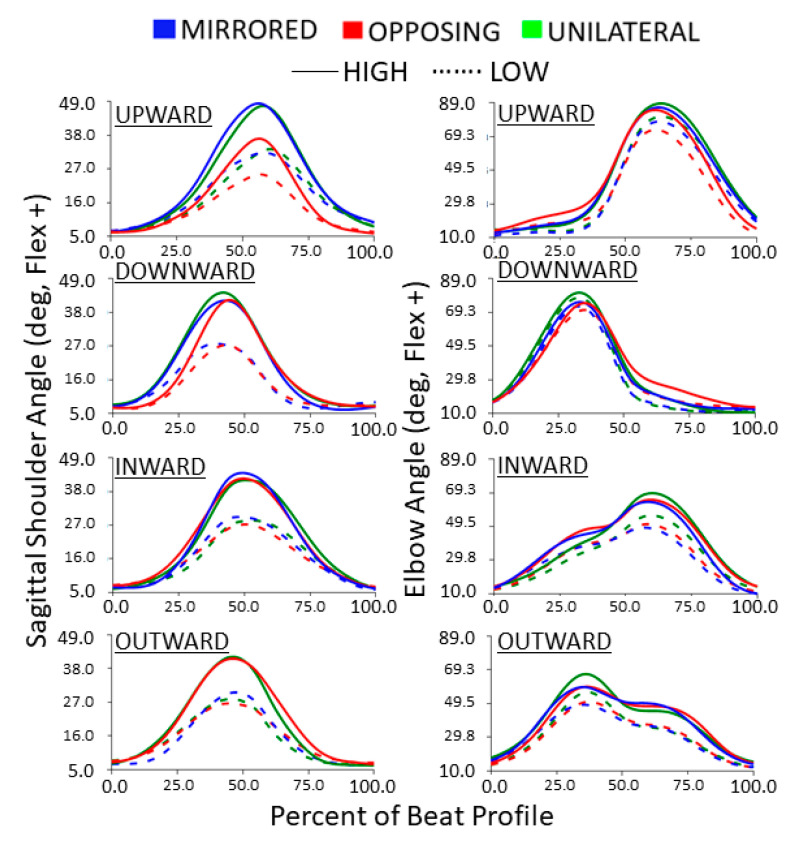
Sagittal shoulder and elbow flexion/extension trajectories normalized to the beat profile. Columns show sagittal shoulder flexion/extension and elbow flexion/extension. Rows show cardinal movement task direction. Movement levels and vertical task position are shown by line color and line style, respectively.

**Figure 9 sensors-24-00233-f009:**
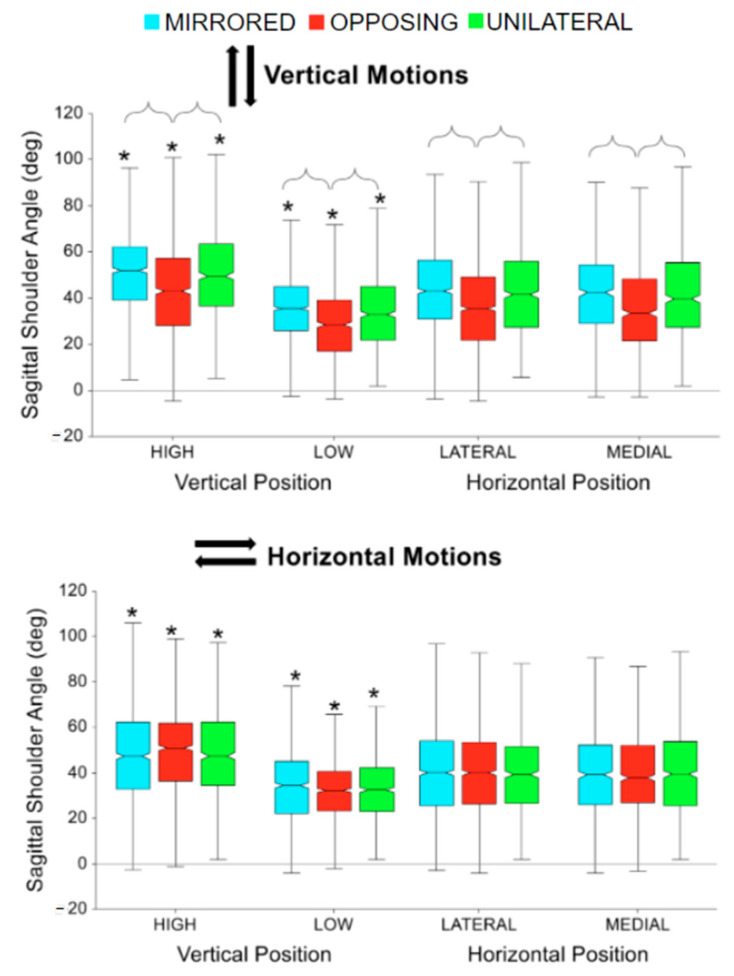
Peak shoulder flexion for vertical and horizontal movements comparing task position and movement level. The * symbol indicates a significant difference between metrics due to the vertical or horizontal position within a movement level, e.g., between unilateral high and unilateral low. Brackets indicate a significant difference between movement levels, e.g., between unilateral high and mirrored high.

**Figure 10 sensors-24-00233-f010:**
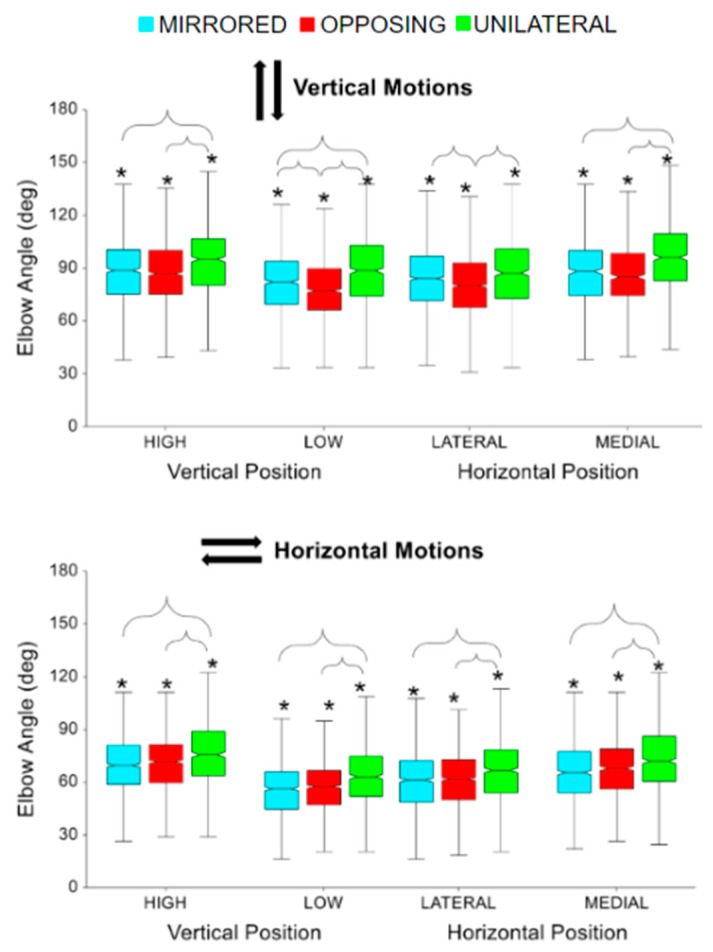
Peak elbow flexion for vertical and horizontal movements comparing task position and movement level. The * symbol indicates a significant difference between metrics due to the vertical or horizontal position within a movement level, e.g., between unilateral high and unilateral low. Brackets indicate a significant difference between movement levels, e.g., between unilateral high and mirrored high.

**Figure 11 sensors-24-00233-f011:**
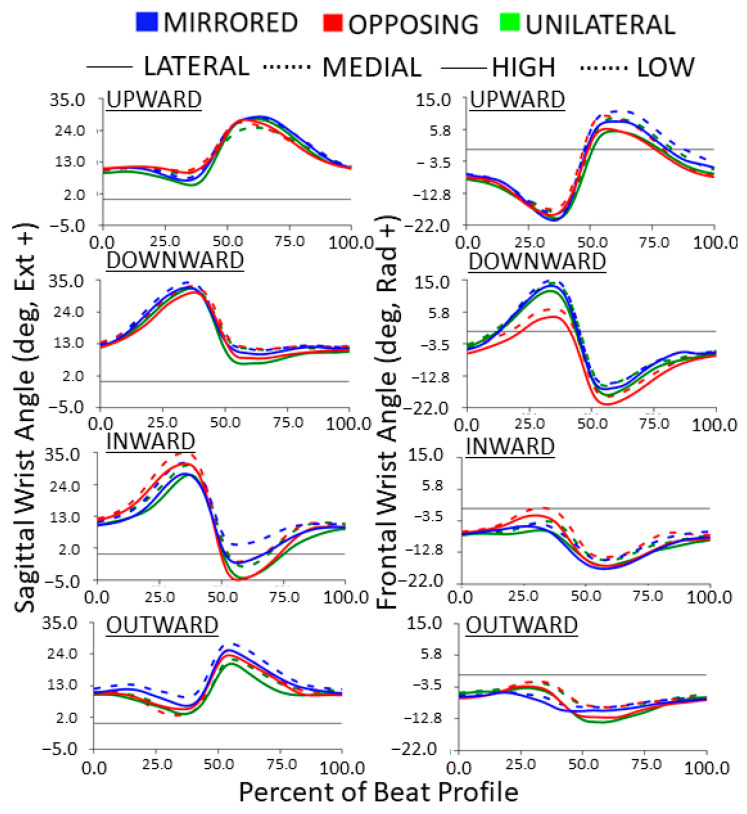
Wrist joint angle trajectories normalized to the beat profile. Columns show sagittal (extension/flexion) and frontal (radial/ulnar deviation) wrist joint angles. Rows show cardinal movement task direction. Movement levels are shown by line color. For the left column, horizontal task position and for the right column, vertical task position, shown by line style.

**Figure 12 sensors-24-00233-f012:**
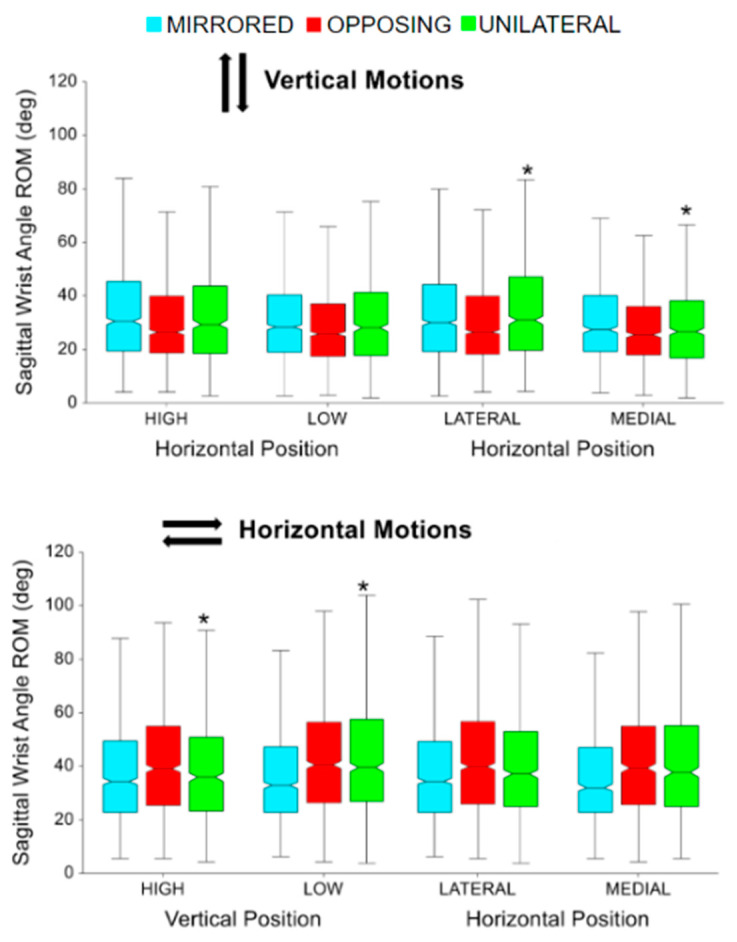
Sagittal wrist angle range of motion (ROM) for vertical and horizontal movements comparing task position and movement level. The * symbol indicates a significant difference between metrics due to the vertical or horizontal position within a movement level, e.g., between unilateral high and unilateral low. Brackets indicate a significant difference between movement levels, e.g., between unilateral high and mirrored high.

**Figure 13 sensors-24-00233-f013:**
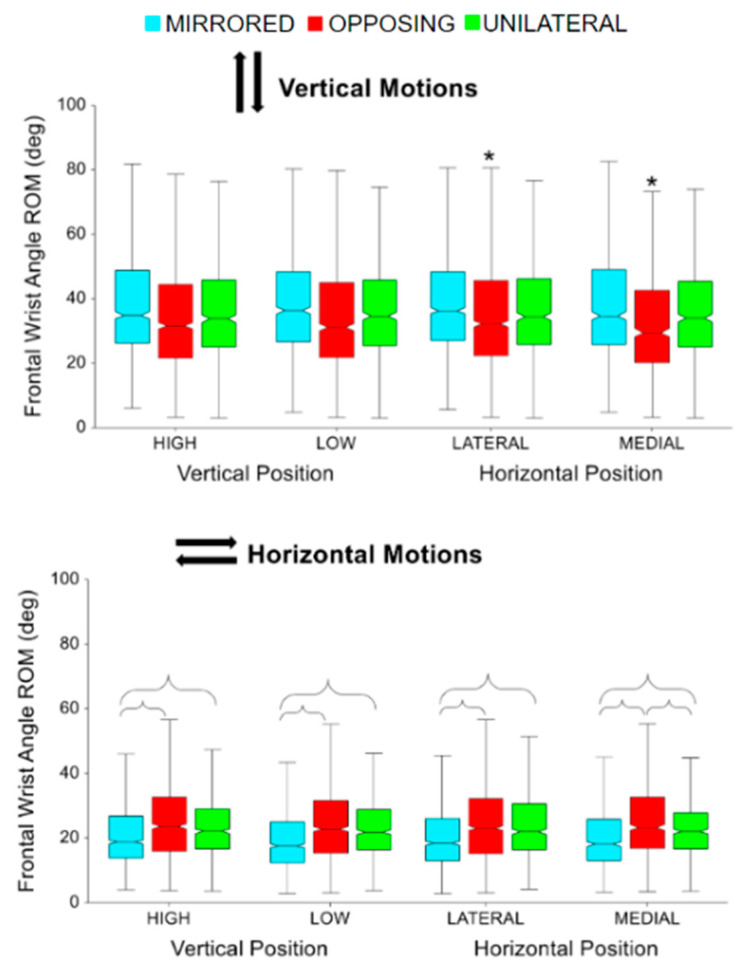
Frontal wrist angle range of motion for vertical and horizontal movements comparing task position and movement level. The * symbol indicates a significant difference between metrics due to the vertical or horizontal position within a movement level, e.g., between unilateral high and unilateral low. Brackets indicate a significant difference between movement levels, e.g., between unilateral high and mirrored high.

**Figure 14 sensors-24-00233-f014:**
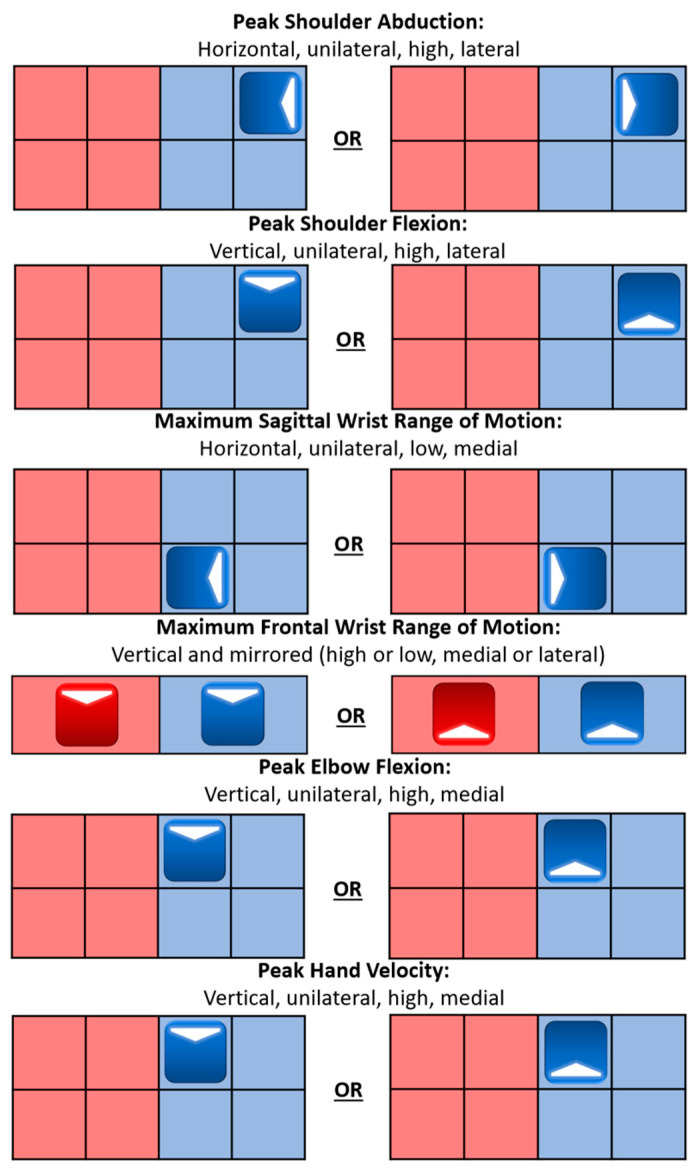
Potential physical therapy guide with each goal and its associated motion: general cardinal movement task direction, movement level, vertical task position, horizontal task position. Red indicates a left arm task and blue indicates a right arm task.

## Data Availability

Data are contained within the article.

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
