# Peer review of "Characterization of Upper Extremity Kinematics Using Virtual Reality Movement Tasks and Wearable IMU Technology"

_sensors, 2023, doi:10.3390/s24010233_

Round 1
Reviewer 1 Report
Comments and Suggestions for Authors
Characterization of Upper Extremity Kinematics Using Virtual Reality Movement Tasks and Wearable IMU Technology
An interesting experimental work that can be improved:
1. The Authors must include an information for the other nine males – kilograms, height, etc.? If the Authors decide it would be better to include a Table with all participants info - weight, height, ....
2. The Abstract and Conclusions can be improved.
Improvement. Based on the proposed corrections and others, these two
paragraphs should be rewritten clearly and in depth. In the Abstract must be
included the main conclusions and the authors must underline the own contributions.
In the Conclusion and in the Abstract it is good, in my opinion, that they emphasize
their contributions and innovations clearly and boldly.
The unwritten rule is that most readers only look at these paragraphs – abstract and conclusions.
3. The References can be enriched with the same from the last 5 years.
Eight of twenty-four references are from the last 5 years 2019 – 2023.
4. In the Introduction the Authors should add a brief summary of similar studies, as well as to
underline their own approach improvements in comparison with the other investigations.
5. A male-female Upper Extremity Kinematics comparison would be interesting to me.
I personally really liked the way the results were presented.
I hope that the proposed corrections will increase the quality of the manuscript and possibly its citability.
Reviewer 2 Report
Comments and Suggestions for Authors
Thank you for inviting me to review this manuscript.
This work investigates VR system usage for locomotor rehabilitation.
This work is well written, chosen topic is perspective. The research design appropriate.
As an addition I would recommend to improve the introduction and add more references.
Moreover, please add some information about next stages of your research
Reviewer 3 Report
Comments and Suggestions for Authors
Thank you for study. The number of subjects in the article is quite small, but I think it will be useful to evaluate Characterization of Upper Extremity Kinematics Using Virtual 2 Reality Movement Tasks and Wearable IMU Technology.
There are a few points that should be added in the Method section.
Inclusion and exclusion criteria can be clearly explained.
In the Discussion section, the results of studies conducted on some pathologies may be included. The recommendations recommended to be applied to patients given in the conclusion section may be supportive. There are many studies in the clinic where virtual reality is used in treatment.
